# Refugee Education under International NGOs: A Major Shift from National Institutions to Patron–Client Relations

Sheraz Akhtar [1,*] and Elizabeth Rata [2]

1 The Faculty of Humanities, Chiang Mai University, Chiang Mai 50200, Thailand
2 The Faculty of Education and Social Work, The University of Auckland, Auckland 1023, New Zealand
* Correspondence: sheraz.a@cmu.ac.th

**Abstract:** What happens when a group of structurally powerless refugees exist within a nation-state's territory but outside its regulatory institutions? An empirical study of the education of Pakistani Christian refugees in Bangkok, Thailand, identifies an entrenched gap between the education provided by INGOs and Pakistani Christian refugee expectations of the academic education of their children. We generalise from the specific problem of the entrenched educational discrepancy to a deeper structural inequality by using a 'realist conceptual methodology' characterised by the type of co-dependency found in the historical form of patron–client relations. The patron–client relationship is the outcome of being placed outside a nation-state's institutions and the co-dependence that the relationship itself creates between the INGO providers and the refugees. We suggest that patron–client theory is a useful conceptual tool with which to explain the sociopolitical position of groups today who find themselves placed outside a modern nation-state's institutions.

**Keywords:** refugee education; INGOs; national institutions; patron–client relations



## 1. Introduction

Patron–client relationships are a historical social form that is widely found in many societies and is used to regulate the power relation between those with resources and status and those without. The emergence of institutions in ancient Greece (Moutsios 2017) and their central place in the modern nation-state, particularly in nations where the institution of the citizen is a key regulator of political relations, has replaced the ancient patron–client form as the main regulator of sociopolitical relationships. What happens when a group of structurally powerless refugees exist within a nation-state's territory but outside its regulatory institutions? This question emerged in the empirical study undertaken by one of us (Akhtar 2020) of the education of Pakistani Christian refugees in Bangkok, Thailand.

The study identified a problematic gap in the education provided for this refugee group by international non-government organizations (INGOs). The INGOs' provision of basic literacy education and the refugees' desire for 'academic education' did not match. The refugees' desire is fuelled by the hope that an education in the academic subjects of the sciences and English, in particular, will enable them to undertake university education. In turn, university-level qualifications are seen to provide the foundation for professional employment, the type of employment that is believed to be the route to citizenship in liberal Western countries and to the shift in status from unwanted refugee to wanted migrant. Despite the refugees' ongoing requests for such an academic education, INGOs continue to provide basic functional literacy in English or in the language of the country where the refugees live in a 'holding pattern'.

How was this ongoing discrepancy between the INGO's provision and the refugees' expectations to be explained? Addressing this question led us to consider the deeper structural relationship between the two parties. On one level, the study of INGOs' education provision in their learning centres in Bangkok enabled us to identify the source of the ongoing gap as one created by Thailand's exclusion policies for refugees. However, this did

not provide a full explanation for the way in which the discrepancy was entrenched. For a deeper sociological explanation, we needed concepts that enabled us to understand the nature of the relationship between the INGOs and the refugees. In turn, that required an analysis of how power works between two parties who operate within a national territory (in this case in Thailand) but who are outside the institutional regulations and services of the state.

## 2. Realist Conceptual Methodology

We use a realist conceptual methodology to theorise the INGO's provision and the refugees' expectations by using key sociological concepts, including 'nation-state', 'patron-client', 'clientelism', and 'academic knowledge'. These concepts operate as a framework to understand the deeper implicit relationship between the two unequal parties and its embodiment in the education gap. Previously, Winter-Villaluz's (2015) study explored how the Pakistani Christian refugees use "social capital to address lack of access to education" (p. v) by using an interpretive approach. However, we are interested in explaining the ongoing discrepancy between the INGO provision and the refugees' expectations. An interpretive approach describes a specific context and is unable to provide a theoretical explanation for the latent structures and procedures affecting refugee communities. For example, the data collected from the participants and other sources cannot reveal the causes, structures, and procedures of the provision of education for the refugees we studied. For this task, we required concepts drawn from the discipline of sociology. This, therefore, leads us to employ a realist methodology (Nola and Sankey 2007). Such a methodology does not draw explanations directly from the empirically obtained data but instead applies theoretical concepts to the empirically obtained data. This approach enables us to move beyond the interpretation offered by the data alone to a conceptually informed analysis and explanation (Nola and Sankey 2007). Applying the concepts in our Thai study justifies our argument that the refugees exist outside the nation-state concept within an older patron–client relation.

Ethical issues were a serious concern in our Thai study because Pakistani Christian refugees are one of the most vulnerable communities in Bangkok and face harassment, discrimination, and exploitation. We approached numerous Thai organisations before undertaking this research since these organisations (such as 'Action International' and 'Rescue Organisation') already have established relations with the community. They are also members of a consortium of organisations called the Thailand Refugee Organisational Network. We used pseudonyms for these organisations in the Thai study to protect the identities of the clients (Akhtar 2020). In order to mitigate the level of influence, we requested the recruiter to emphasise the voluntary nature of our study. Prior to interviewing the research participants, one of the authors ensured that the refugees understood the content form before signing it.

In order to build an argument that would explain the entrenchment of the gap between the INGO provision of education and the refugees' expectations, we investigated how such extra-state provider–recipient relations operated historically. This historical analysis showed the role of 'patron-client' relations as a significant relationship in regulating the power imbalance between individuals and groups in older pre-modern times. It is a relationship characterised by differential power in terms of the provision and receipt of resources (Freedman 2015; Roniger 1983, 2015; Scott 1972a; Stein 1984; Eisenstadt and Roniger 1980; Galt 1974; Hall 1974; Foster 1963).

The final stage was to generalise the patron–client concept to explaining the entrenchment of the power imbalance between the INGOs and the refugees—an ongoing imbalance that manifested itself in the entrenched gap between the type of education provided to the Pakistani Christian refugees and their expectations for an academic education for their children.

Patron–client theory and the mechanism of the 'clientelistic tool' can be justified as an explanatory tool for excavating and explaining the problem for two reasons. First,

they allow for the recognition of a less structured exchange of goods and services that characterises the weaker, less institutional solidarity (Eisenstadt and Roniger 1980) between powerful INGO providers and structurally powerless refugees. This is in contrast to the stronger institutional solidarity found in the contemporary nation-state system, where laws and policies regulate the relationships between parties. Second, the patron–client concept was useful for the empirical study about the Pakistani Christian refugees in Thailand specifically. Thailand legally permits refugee children to have access to primary and secondary education in public schools, but "implementing this strong legal framework has not been successful" (Save the Children n.d.). This is because discrimination, xenophobia, and the long distances from schools without transportation support leave the refugee children to enrol at community learning centres established by INGOs. The patron–client theory enabled us to theorise the study's findings of an entrenched gap between education provision and refugee expectations in terms of an unequal power relationship. This is a structural inequality characterised by the type of co-dependency found in the patron–client relation. We were particularly interested in how the structuring of that co-dependency served to maintain the lacunae between the INGOs' education provision and the refugees' educational expectations.

Of particular interest was the way in which the relationship between the two parties demonstrated the features of the older patron–client relation identified in the historical analysis. Despite the refugees viewing their time in Thailand as temporary, the co-dependence of provider and recipient creates a structural relationship that enables a permanent stasis. Unless the Pakistani Christian refugees return to their country of origin, there is no reason why this arrangement cannot continue indefinitely. It is external to the Thai state; it provides benefits to the NGOs by ensuring their continued existence and funding; and it provides some, albeit very limited, resources to the refugees. The refugee route to Western countries, which existed in the post-World War Two decades, no longer exists as an alternative to this stasis. Therefore, we ask: is this older patron–client relationship one that will define the refugee experience in the foreseeable future?

If this is the case, then refugees will remain stuck in a permanent limbo. The emerging countries to which they often initially escape exclude them from state provisions, and developed nations restrict access (Mertus 1998; Chimni 2004; Betts and Collier 2017). Instead, an older relationship appears to have resurfaced: that of the INGO 'patron' and the refugee 'client'. This is an older co-dependency relationship without the accountability and regulatory systems available to those who are recognised and practised by nation-states (Najam 1996).

Before discussing the second and the third paths, the unlikely options, we use a historical analysis to explore the reasons why the provider–recipient (INGOs–refugee) relations in the contemporary world exist outside of the nation-state system.

### 3. Inclusion: The History of Refugees

The historical analysis of refugees from the 20th to the 21st centuries enables us to locate the status of displaced people as either 'migrants' or 'refugees' in the context of changes to global capitalism's historical management of labour. Displaced people were first unwanted by liberal Western countries (resettlement countries) in the post-First World War period, but later they were welcomed by these advanced countries in their nation-state system in the post-Second World War period. However, Jewish refugees had a difficult time integrating into Western Christian countries, and they were labelled as 'enemy aliens' (Beaglehole 1988). The high labour demand in the West from 1945 to 1970 began an 'open door policy' for refugees (Chimni 2004; Hollifield 2004; Suhrke and Newland 2001). They provided a much-required labour force to rebuild Western European countries' infrastructure (Miller and Martin 1982). That initiated 'a golden era' (Suhrke and Newland 2001, p. 285) for refugees to become desirable migrants and to secure citizenship in Western countries.

Our historical analysis included the historical refugee situation in the South Asian region where the Pakistani Christian refugees in the Thai study are located. The religious identification is significant for historical reasons. The British India partition in 1947 was founded on religious distinctions between Hindus and Muslims, with millions of refugees' moving to the newly formed nations of India and Pakistan. Their inclusion in the nation-state systems of these countries was based on distinct religious identities, with Muslims seeking refuge in Pakistan and Hindus seeking refuge in India (Talbot 2011).

The refugees in this period contributed significantly to the socioeconomic development of these nations in the modernisation of the economies (Oberoi 2005). This is why both countries took a number of strategic measures to include the refugees in their labour market (Gabriel 2013). By doing so, the host countries ensured the refugees would not become a burden on their inadequate socioeconomic resources but rather contribute to their economy. For example, Pakistan set up the Refugee Finance Corporation for loans and grants. Vocational and technical training centres and higher education scholarship programmes were established to ensure refugees' inclusion in the labour market (Oberoi 2005; Peshkin 1963). In this way, the refugees' inclusion in both countries was within the nation-building tradition of ensuring that arrivals became migrant workers that were subject to the labour management policies of the nation and were able to access social resources, including education.

The 1970s, however, was a time of economic decline and rising unemployment rates in Western nations (Siebert 1997). Western governments struggled to offer labour opportunities to their citizens, and the massive influx of unskilled refugees added extra pressure to the labour market. Developed nations changed their immigration policies from accommodating unskilled foreign workers, including refugees, to welcoming more limited numbers of skilled workers who could contribute to a host country's economy (Suhrke and Newland 2001; Hollifield 2004). These policies were intended to secure borders that would control refugee movement. Sanctions were placed on airlines for transporting refugees, and strict visa policies were initiated for the refugee-generating countries (Suhrke and Newland 2001; Collinson 1996; Betts and Collier 2017; Tropey 2000).

## 4. Exclusion: Contemporary Refugee Context

The major policy changes in the 1970s created the contemporary, post-1980s global context for refugees. These decades are characterised by two features. First, people who leave their country of origin to escape persecution often find themselves in neighbouring emerging countries or pre-resettlement countries (Betts and Collier 2017), something they regard as a temporary stage before being accepted by an advanced country.

The Pakistani Christian refugees in Bangkok escaped Pakistan to avoid political exclusion (Rais 2005), blasphemy laws (Rahman 2012), religious radicalization (Thames 2014), and inadequate protection from the state's institutions (Rahman 2012). As with refugees in other countries, these people believed that their stay in Thailand was to be temporary. However, this is not the case. Many refugees have been living in the city for more than nine years and are still waiting for their resettlement to liberal Western countries, a resettlement that is increasingly unlikely. The Pakistani Christian refugees in Thailand, like many others (for example, millions of Afghan refugees in Pakistan, Syrian refugees in Turkey, and Rohingya refugees in Bangladesh), are now restricted to pre-resettlement countries as a consequence of the post-1970s global economic, political, and cultural context.

The second feature of these decades is the significant policy shifts regarding refugee status that were implemented as repatriation and other restrictive measures came to characterise the contemporary global refugee situation (Mertus 1998; Chimni 2004). These policy shifts can be explained in terms of the limited numbers of refugees accepted by developed nations, leaving many others stuck in pre-resettlement countries. Growing refugee populations in the latter countries place considerable burdens on the limited social services. Betts and Collier noted that:

*With democratization, debt crises, and the 'Structural Adjustment' programme of the 1980s and 1990s through which the International Monetary Fund and the World Bank imposed economic liberalization and cuts in government spending across much of the developing world, host governments became increasingly constrained in their ability to allocate scarce resources to non-citizens.* (p. 41)

Emerging countries such as Thailand face financial challenges in delivering social services to their own citizens. Additionally, these countries are not signatories of the Refugee Convention of 1951 and its protocol in 1967, and therefore they are not legally obligated to ensure refugees' legal, social, and economic rights. As a consequence, refugees are excluded from being recognised by national institutions in emerging countries. Conversely, as signatory members, advanced countries are legally required by the Refugee Convention to include refugees in their social institutions and even provide a clear path for them to become citizens. In Thailand, the institutions whose policies exclude refugees include the Department of Social Development and Welfare, the Ministry of Public Health, the National Democratic Institute, and the Ministry of Labour. INGOs fill the gap left by the absence of state services. However, the services provided by the INGOs are often limited (Betts and Collier 2017).

## 5. Accountability

INGOs are structurally differently from government departments and operate outside nation-state regulations. This is a significant feature for our 'patron-client' thesis and is discussed in detail below. Waters and Leblanc (2005, p. 132) suggested that INGOs function as "pseudo state" in emerging countries, leading to questions regarding the INGOs' accountability process in those countries (Najam 1996). Najam (1996, p. 340) argued that the INGOs' accountability has been "confused with much narrower and short-term concepts of projects evaluation and monitoring" processes. His study categorised INGOs' accountability into two forms: "functional accountability (accounting for resources, resources use, and immediate impacts) and strategic accountability (accounting for impacts that an INGOs' actions have on the actions of other organisations and the wider community)" (Najam 1996, p. 351).

The study's findings showed that INGOs have high functional and medium strategic accountability to governments and funding agencies. They have low to nil functional and strategic accountability to recipients, including refugees. The low to nil accountability indicates that INGO providers and recipients' relations are neither bound by nor accountable to nation-state regulations (Akhtar 2020). There are considerable implications that follow from this. The Thai study investigated how these implications are experienced in education by focussing on the education gap between the refugee parents' aspirations and the INGOs' provisions. The aim was to demonstrate what happens for refugees who are stuck in this extra-national relationship.

## 6. Patron–Client Theory

In order to explain this INGO–refugee relationship in terms of a group situated outside the nation's political, legal, and social institutions but actually living within the nation's territory, we turned to patron–client theory (also referred to as 'clientelism'). We justify its use for two reasons. The theory can be applied to a specific refugee group, as we use it to explain the circumstances of the Pakistani Christians in Thailand (Scott 1972a; Roniger 2001). Second, 'patron-client' serves a generalising function. The theory can be used to explain how the refugee movement is organised globally (Akhtar 2020). It enables an explanation of the lopsided power relationship characterised by co-dependency more generally in INGO–refugee relations and provides a tool to explain the educational entrenchment in the community learning centres in Bangkok. (These centres were the sites for the empirical study of the Pakistani Christian refugees in Thailand.)

Given the usefulness of patron–client theory, we discuss it in detail in this section. First, we define the patron–client relationship and describe its types, structures, and an

exchange mechanism called 'patronage' or 'clientelism' (interchangeable terms). Second, we argue that these relations have existed historically and, crucially for our purposes, that they can be seen today in INGO–refugee relations, including those found in the Thai study, but they are also generalisable to refugees in other countries. Scott's (1972a) account of the patron–client relationship provides a useful definition:

> *The patron-client relationship–an exchange relationship between roles–may be defined as a special case of dyadic (two-person) ties involving a largely instrumental friendship in which an individual of higher socioeconomic status (patron) uses his own influence and resources to provide protection or benefits, or both, for a person of lower status (client) who, for his part, reciprocates by offering general support and assistance, including personal services, to the patron.* (p. 92 italics in the original text)

The exchange relationship identified by Scott (1972b) recognises well-defined roles between patrons and clients in rural Southeast Asia. Subsequently, Roniger's (2001, 2015) study built on Scott's research and further theorised how the patron–client relations persists, in one way or another, in complex contemporary societies. Fundamentally, the patrons' role is to protect clients from other influential patrons and provide access to resources in challenging times such as illnesses, famines, floods, and storms. In exchange, clients are required to offer personal services such as labour in fields or factotums and show allegiance and compliance to their patron(s) (Wallace-Hadrill 1989). A three-party complex exchange involves an additional actor called a 'broker' or a middle person whose role is to connect a patron to clients, but the broker "does not himself control the thing transfered" (Scott 1972a, p. 95).

In patron–client relations, the patrons' hierarchical and influential position in the movement of resources from patrons to clients provides greater advantages to the former. The advantages include building their public reputation and broadening their influence in society (Scott 1972b). In contrast, clients may receive some benefits in emergency situations, but their lower standing in the relationship puts them in a disadvantageous position (Galt 1974; Stein 1984). For refugees specifically, the main disadvantage is their entrenchment in an ongoing and unequal relationship to the patron—the INGO. We return to this discussion of 'entrenchment' below, with reference to the specific case of refugee education in Bangkok, but first we describe patron–client types, structures, clientelism, and the historical origins of this relationship.

## 7. Patron–Client Types

We observed that INGOs and a few refugee leaders were directly connected to each other through the dyadic relations identified by Foster (1963) and Scott (1972a, 1972b). The refugees with higher education and fluent English language are often the ones who can establish these direct relations with INGOs, relations that enable access to scarce resources. Afterward, the refugees play a mediator role in connecting the refugee community to the INGOs. The mediator role of brokers extends the dyadic relations into a triadic form (Scott 1972b). For example, one of the refugees who participated in our study said, *"if we need help, we usually ask our Pakistani brothers to share our needs with INGO leadership"* (Akhtar 2020, p. 111). The 'brothers' were the brokers. In this way, the refugee community leaders who serve as brokers concomitantly act both as clients and refugees at a specific time (see Figures 1 and 2). Taking the broker role, in addition to being a client, enhances the refugee leaders' position in the triadic relations, and that position provides them with favours from INGOs. The favours enable sufficient social provision and include food rations, health benefits, education, and salaries.

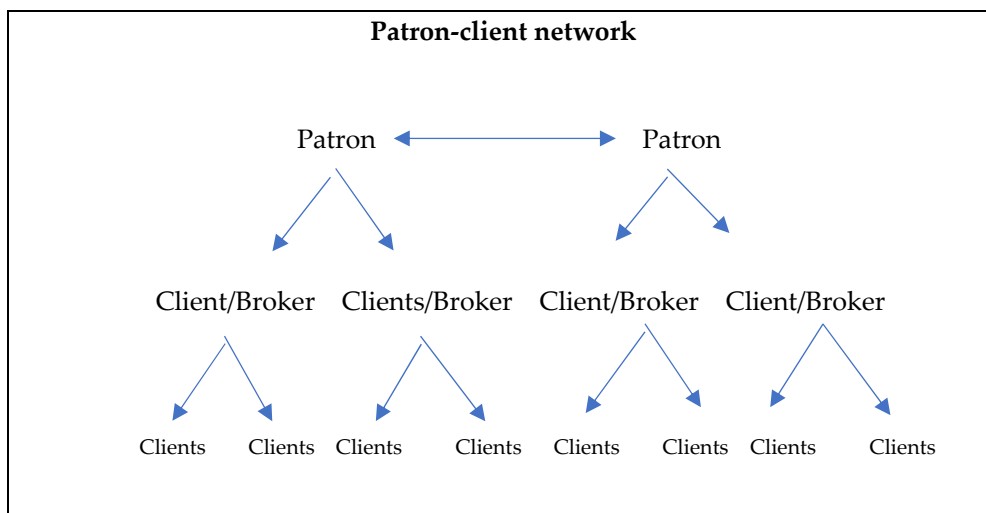

**Figure 1.** Patron-client network.

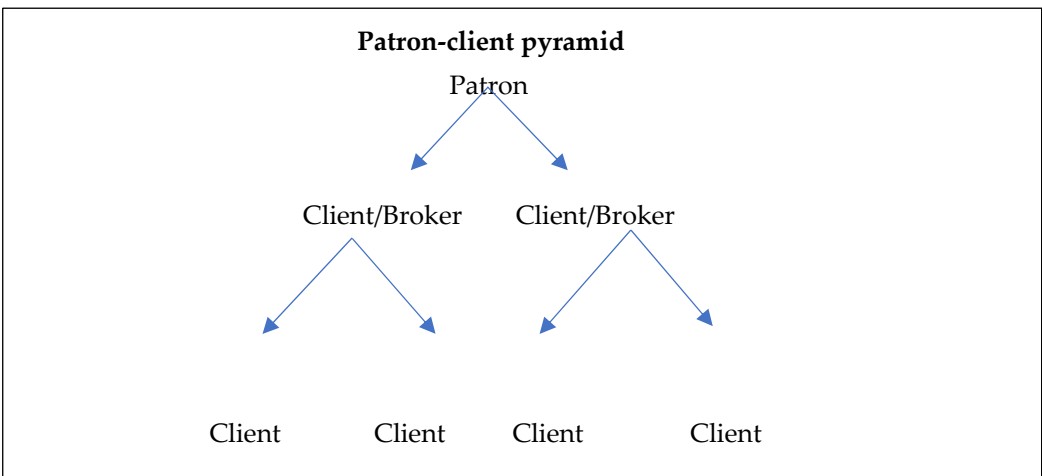

**Figure 2.** Patron-client pyramid.

## 8. Patron–Client Structures

Scott (1972a) identifies three types of patron–client structures: cluster, pyramid, and network. The patron–client cluster is the first structure in which a patron is directly connected to many clients (see Figure 3). This structure is established in traditionally homogenous religious, ethnic, or caste groups (Scott 1972b), but in modern society this cluster may be built on occupation or social status (Saxebol 2002; Roniger 2015), particularly when established between non-kin groups. The second structure, the patron–client pyramid, is a vertical downward extension of the patron–client cluster but still includes one patron and many clients. Both the cluster and pyramid structures contain a vertical and hierarchical relationship between a patron and clients. This downward extension in the patron–client pyramid is usually initiated by the reciprocal agreement between a patron and those clients who operate as brokers. They play a key role in introducing new clients to multiple patrons (see Figures 1 and 2).

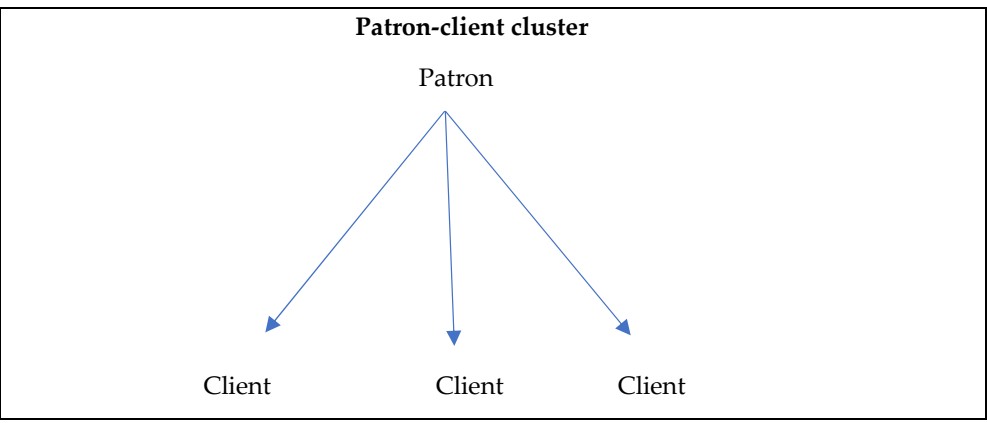

**Figure 3.** Patron-client cluster.

The third network structure, or the patron–client network, involves two or more patron–client pyramids in which patrons cooperate with each other for their own benefits (Roniger 2015). This structure encompasses horizontal and vertical relationships between numerous patrons, brokers, and clients. Patrons with close socioeconomic statuses establish horizontal relationships (Roniger 2015) in addition to the vertical hierarchical relationships of the cluster and pyramid structures, which are often found in societies based on ascriptive hereditary status with large differences in socioeconomic and political status (Abercrombie and Hill 1976). The horizontal structures of the network type, in contrast, are mostly created among patrons with less socioeconomic and political status differences and consequently offer a slightly more balanced power structure (see Figure 1). In the vertical structure, a type found mainly in non-democratic societies with weak governance and hereditary status relations, patrons hold power and control over their clients (Abercrombie and Hill 1976). However, in all three structures identified by Scott (1972a) the patron–client relationship is a power relationship where patrons retain their authority and power over social, political, legal, and economic resources.

**9. INGO–Refugee Structures**

The contemporary refugee situation in Bangkok may be considered as the network type because, first, many INGO providers are involved in the provision of social services with specific provisions. For example, the Action International and Rescue Organisation only provides education to the refugee community. Second, the Pakistani Christian refugee community does not have direct access to various INGOs. This means they may require multiple brokers for different resources such as education, health, and food rations because each INGO has its own set of brokers. Neher (1994) reported that "[i]n urban areas, patron-client ties are more specialised and impersonal, and a client will arrange to have more than one patron so as to meet multiple needs" (p. 950).

INGOs in Bangkok fit Roniger's (2015) explanation of a complex patron–client network (see Figure 1) in contemporary societies. The key feature is patrons cooperating with each other for their own benefits. This can be seen in, for example, the United Nations High Commissioner for Refugees (UNHCR) and its partner organisations such as the 'Action International and Rescue Organisation' having regular monthly meetings to decide about the provision of social services to the Pakistani Christian refugees. The 'Action International and Rescue Organisation' and other organisations investigated in the Thai study were a part of this network. The refugee participants in the study said that the INGOs do not invite them to those meetings. The vertical hierarchical relationship between the INGOs and the Pakistani Christian refugees in Bangkok was also identified by Waters and Leblanc (2005, p. 130). Their study also found that "planning is often done 'for' refugees by external actors like the host country, United Nations (UN) relief agencies, and nongovernment organisations, rather than 'with' refugees".

## 10. Patron–Client Historical Origins

The type of power relationship theorised as a patron–client one is well-documented historically to have existed in different times and spaces in various societies with weak governance and less socioeconomic and political status. We argue that throughout history societies had hierarchies of power, especially for people with less advantageous backgrounds, such as refugees, who require assistance from powerful segments of society in order to gain stability and much needed resources. This system was found in the early Roman period (Wallace-Hadrill 1989) and in ancient Greek societies (Gallant 1991). By the European Middle Ages, John of Salisbury (1115–1180), a political thinker, recorded that the patron–client system was celebrated during the papacy of Adrian IV (1100–1159 AD). Later, the principle of patron–client relations was found in the system of servitude in Europe, commonly known as serfdom or feudalism (Scott and Marshall 2009). Another example of patron–client relations could be seen in *mezzadria* system developed between landowners and tenants in central Italy (Silverman 1965). Sociologists and anthropologists also documented the characteristics of patron–client relations in the *compadrazgo*, or godparenting, system in Latin America. Likewise, hierarchies of power also existed in the *biraderi* (brotherhood) system in the Pakistani society, from ancient times to present, where the refugee participants of our study come from.

A similar story of foreign migrants attaching themselves to a powerful patron in order to acquire at least a degree of stability and access to resources in the new and often unwelcoming land was repeated two millennia later. The Thai study showed how refugees sought the provision of social services and protection from INGOs in the difficult and hostile conditions they frequently faced in Bangkok. In return, the INGOs' staff used the refugees for fundraising activities and household chores such as babysitting, gardening, and cooking, thereby beginning the co-dependency at a very personal level.

INGOs claim to provide sufficient social services, including education and protection. All refugee participants in our study, however, showed dissatisfaction with the limited provision of these social services and the unequal nature of the relationship, as shown by the different resources available to the two parties. In a comparison to Roman patron–client relations, Cloud (1989, p. 206) noted that Roman patrons usually provided "their clients with food of a lower standard than that eaten by themselves". Likewise, the refugee participants in our study said:

> *"A refugee organisation only provides five kilograms of atta (wheat flour), daal (lentils), one kilogram of cooking oil, and six packets of two hundred millilitres of milk for our family of five on a monthly basis".* (Akhtar 2020, p. 125)

The central role of institutions in removing the political relation from direct personal connections was described by Moutsios (2017) as the cornerstone of democracy in Ancient Greece. These institutions did not accommodate foreigners who had migrated to Athens. As a consequence, the patron–client relations only existed between citizens and these migrant foreigners (Arnaoutoglou 1994). It is analogous to the current situation in which today's refugees find themselves. They too are excluded from the institutions in the emerging countries to which they first fled. The refugees in our Thai study exemplify this. Without citizenship rights, they must seek protection from outside the institutions of the nation-state, leaving them no choice but to rely on the INGOs' provisions, including that of education for their children.

## 11. *Biraderi* System

While the Ancient Greece example is useful for the importance it gives to the role of institutions in replacing patron–client relations as the regulating political relationships, the 'biraderi' (brotherhood) system is also illustrative of a type of patron–client relationship. Its familiarity to the Pakistani Christian refugees, many of whom are from the West Punjab region of Pakistan justifies our interest in it here. Like the older form of patron–client relations, biraderi has the key features of hierarchical, invisible, and unaccountable power structures and co-dependency between the parties.

Although there are distinct religious identities between Christians and Muslims in the West Punjab region, they both share similar social structures (Ismail 1983). The *biraderi* system usually offers "security and shelter" from a hostile environment. However, many Pakistani Christian communities do not benefit from the system because of corruption, which is a deep-rooted issue in Pakistani society (Javid 2010) and often excludes these communities from access to social resources. As a result, some Pakistani Christians are forced to develop other patron–client relations outside of their *biraderi* system to receive favours and social services (Javid 2010).

After the partition of the subcontinent in 1947, the Pakistani government allotted land for the millions of refugees, but *zamindars* (landlords or patrons) confiscated vulnerable refugees' land. Additionally, 'Tahsildars [district officers] and other revenue officers were openly taking bribes from the refugees" (Chattha 2012, p. 1195). This inhospitable environment led the refugees to seek protection from local Punjabi patrons. The patron–client relations existed among Muslims, but Christians were also involved in the system. According to McClintock (1992), patron–client relations also exist within the Pakistani Christian institutions. Pakistani Christian leaders (patrons) often use their authority to provide resources and employment opportunities to their clients. In response, clients are expected to provide "active support in any power struggle" (McClintock 1992, p. 352) within the institutions. This brief account shows the extent to which Pakistani Christians are familiar with patron–client relations in their country of origin.

Having fled to Thailand, the Pakistani Christian refugees seek to establish relationships with INGOs, given that access to social services, including education, is available only through INGO provisions (Akhtar 2020). The refugee children cannot undertake the two-hour commute to the Bangkok Refugee Centre, nor can refugee families afford the high tuition fees at international schools. This leaves the community learning centres, operated by the Action International and Rescue Organisation we studied, as the only viable option. It was in the community learning centres that we identified the way in which a co-dependence operated as the key exchange mechanism between the INGOs and the recipients. This led to our use of the terms 'clientelism' or 'patronage' to explain the fixed nature of the co-dependency. The INGOs needed the refugees in order for the former to secure and justify funding for the centre. The refugees needed the INGOs because they were not entitled to any viable government-provided education. This mutual co-dependency, which becomes structured despite being unequal, is theorised in greater detail in the next section.

## 12. Clientelism

Belshaw (1965) notes that the different types of social exchange hold a substantial value in tying societies together. The scarcity of resources to vulnerable clients leads to an exchange of goods and services, or clientelism, between the patrons who control access to the resources and the clients desperately requiring those resources.

In our study, education can be understood as a type of clientelistic exchange between the INGOs and the refugees. The refugee community desires an academic education for their children, but the INGOs have the control and power to only provide basic literacy education. We argue that an education gap, which is deeply rooted in the unequal power relationships, can be theorised as a clientelistic exchange. It simultaneously contains two types of contemporaneous but different forms of social exchange: 'specific' (instrument or market-like) and 'generalised' (expressive) exchange (Homans 1958; Blau 1964).

A specific exchange is a market-like exchange where social actors have reciprocal interests and agreements about terms and conditions before the exchange of goods and services (Homans 1958; Blau 1964). The anthropologist Marcel Mauss (1872–1950) coined the term 'general exchange' to distinguish a gift exchange from a market-like exchange. In contrast to a market exchange, a gift exchange does not contain any contract, agreement, or time framework, despite the mutual understanding that exists between the patron and the client. According to Sahlins (1965), a gift exchange does not stipulate a material exchange

or impose specific timings on the return. Instead, it is a generalised exchange, often based on expressive values such as a promise of loyalty, trust, and obedience.

The INGO–refugee relationships in the Bangkok community learning centres can be understood using both specific and generalised exchange (Roniger 1983). As a specific exchange, the INGOs maximise their reward by providing social services requiring low operational costs to refugees but expecting the refugees to be involved in fundraising activities. These usually require the refugees to take part in photos, videos, and stories that are used by the INGOs to obtain funding from donors in developed nations. Despite the inadequacy of the educational resources provided to the refugees, the INGOs are able to maintain and even increase their reputation and funds. The refugees, on the other hand, desire to maximise the quality and quantity of social services, including education, but do not have direct access to the funders. For this, the brokerage role of the INGOs is required.

The refugee community in the Thai study wanted their children to receive an academic education in exchange for their services. For them, the education exchange was a specific exchange. In contrast, the INGOs view the exchange of education as a generalised exchange where they are providing basic literacy education in response to an emergency situation, despite the 'emergency' being ongoing with no end in sight. All the interviewed INGO employees and volunteer teachers said that they were only willing to provide basic literacy education through the community learning centres. This is because, first, this type of education helps the INGOs to minimise their operational costs by using uncertified teachers who teach only a few hours per week rather than hiring certified teachers.

Second, a humanitarian aid worker justified the INGO position for the provision of basic educating by noting, *"if we provide them proper academic education, they are not going to leave from here"* (Akhtar 2020, p. 134). This supports the view that the INGOs' are committed to implementing an involuntary repatriation policy as a durable solution for refugee crises (Chimni 2004; Mertus 1998). It requires maintaining the 'emergency' state of the refugees despite the length of the time spent in Thailand and the fact that children have been born in that country. The hope of the Pakistani Christian refugees for their children continues to be acquiring an academic education. This is seen as the only path from unwelcome refugee to desirable migrant professional and the only route to citizenship in a Western nation.

## 13. Patron–Client Contradictions

The clientelistic exchange in education identified in the Thai study concomitantly involves friendship and hierarchical relationships between the INGOs and the refugees. It is a co-dependency characterised by a high degree of personalised relations. All our study participants, including the INGO employees and the refugees, used familial language, often referring to one another brothers and sisters. However, the kin terms appeared to be symbolic only, as one of the refugee parents said:

> *They [INGO employee] deliver us food rations and teach our children, but most of the time they do not talk to us. I do not think that they even know our names.* (Akhtar 2020, p. 134)

The INGO employees were from Western societies and were unfamiliar with Pakistani sociocultural norms and the use of *Urdu* (Pakistani national language). This was despite providing education and other social services to the refugee community for more than nine years. It is likely that the UNHCR's repatriation policies (see above) restricts them from investing time and effort in engaging with the refugee community. A humanitarian worker in the study used an insightful metaphor to explain why these two social actors do not socialise with each other. He said, *"there is an [socio]economic gap between the INGO and the refugees. It is more like water and oil, it does not mix together"* (Akhtar 2020, p. 134).

The refugee parents continued to request that the community learning centres provide their children with an academic education. However, the untrained volunteer teachers continued to ignore their expectations and, seeing themselves as part-timers, would only provide basic literacy education on an emergency basis. An INGO employee noted in a

comment about the mainly part-time volunteer teachers, *"They do not know who the refugees are . . . They do not care about their [the refugees] educational aspirations"* (Akhtar 2020, p. 135).

The provision of basic literacy education through the community learning centres provides an ongoing advantage for the INGOs. The approach has low operational costs, with community learning centres operating in one room and with volunteer teachers. The INGOs continue to promote the importance of basic literacy education for refugees despite the fact that basic literacy education locks the refugee children into a permanent refugee state. A refugee parent commented on being stuck in this entrenched position:

> *We cannot afford international schools and have limited options so, our children have to learn the basic literacy education in the community learning centre.* (Akhtar 2020, p. 136)

The refugees have no other option than relying on the INGOs' provision of basic education. It is a situation confused by the inherent contradictions of the patron–client relation, in which friendship and familial symbols conflict with the unequal hierarchies of the INGO patron and the refugee client (Roniger 2015; Scott 1972b). The advantages to both parties, although significantly less to the refugees, serve to maintain this relationship without any means to enact change.

The contradictions inherent to the patron–client relationship can be seen in the use of favouritism. Although patrons claim to show benevolence to their clients, they demonstrate favouritism towards brokers from the refugee community who play a vital role in connecting the two social actors (Akhtar 2020). For example, most refugee families in our study received inadequate provisions of food rations and shelter, and the basic literacy provided was not what the refugees wanted. In contrast, the two broker refugee families had their children enrolled in the international schools on full scholarships. One of the refugee families said, *"we are so thankful for our friends for giving us full tuition fee scholarship for our children"* (Akhtar 2020, p. 138). However, a very different picture was provided by an INGO employee:

> *The reason those children are in the international schools because their parents do not have passive behaviour as others. The parents constantly seek new opportunities and ask their friends for assistance.* (Akhtar 2020, p. 138)

## 14. Power-Structured Relationship

The presence of ongoing inequalities in patron–client relations is well-documented in the literature regarding power-structures between powerful patrons and powerless clients (Roniger 1983, 2015; Scott 1972a; Galt 1974; Abercrombie and Hill 1976). Some argue that this relation may alleviate inequalities (Eisenstadt and Roniger 1980). However, Galt (1974), Stein (1984), and (Akhtar 2020) note that, while the patron–client relationship may provide stability in a short-term emergency situation, the relationship perpetuates inequalities in the long-term. Despite the appearance that the relationship is beneficial for clients, it is founded on "inequality and hierarchy" (Roniger 2015, p. 604). These ongoing inequalities structure differential power into the patron–client relationship (Stein 1984). Patrons have control over alleviating or maintaining inequalities.

It was clear from the empirical data obtained in the study of the Pakistani Christian refugees in Bangkok that the patron–client relationship was of the vertical type (Abercrombie and Hill 1976). This was most noticeably demonstrated by the fact that the refugees have no representation in decision-making meetings. The engagement between the two parties takes the form of flexible transactions and is characterised by the absence of a signed agreement. In contrast, the consortium of INGOs, including the organisation we studied, operates according to a signed Memorandum of Understanding establishing the horizontal structure of the INGOs relations (Abercrombie and Hill 1976). Significantly, this memorandum serves as an accountability tool—something absent from the INGO–refugee relationship.

Although the refugees view their time in Thailand as temporary, their exclusion from the national institutions means that they have little choice but to rely on the INGOs for

protection and social services, including education. They have limited access to the Thai public school system and do not have the authorisation to work in Thailand. Given this position, it is unsurprising that the refugees are totally reliant on the INGOs. One of the refugees who took part in the study said succinctly, *"we cannot survive here [in Bangkok] without the INGO provision"* (Akhtar 2020, p. 137).

Greater benefits accrue to the INGOs. They need the cooperation of the refugees to secure future project funds, so to some extent they are in both an advantaged position as well as one with a degree of dependency on the supplicant party. However, the refugees are without a doubt the disadvantaged party in this co-dependent relationship (Scott 1972a). They have no other options except to accept the basic literacy education that is offered.

In the long term, their options are severely limited, with only two options available, neither of which is desirable. The first option is that the refugees will repatriate and reconnect to their national institutions, including the Pakistani public education system. Despite this being the option rejected by the refugees, it is the preferred policy of the donor nations and one endorsed by the UNHCR. The second and more unlikely option is that the refugees will be permitted to resettle in one of the liberal Western countries. However, Western countries only accept one to two percent of the worldwide refugee population (Betts and Collier 2017). Their policies are designed to maintain refugee populations in the country of first arrival, which is usually an emerging country, as is the case with the Pakistani Christian refugees' placement in Thailand (Betts and Collier 2017).

Most refugees decide to live in emerging countries and wait for resettlement in the Western countries—a highly unlikely scenario for the contemporary refugee situation. Their hope of receiving an academic education is directly related to these resettlement aspirations. Despite their often decades-long stay in the first country of arrival, the refugees continue to exist outside of the nation-state system (Waters and Leblanc 2005), many for more than twenty years, with an ongoing dependency on the INGOs (Betts and Collier 2017). It is this 'temporary but permanent' situation that has reactivated the older patron–client relationship.

## 15. Refugee Education

Given these circumstances, is there any hope that the refugee parents' aspirations for academic education can be met? It is a question that requires a discussion of what academic education is. With reference to realist educationalists (e.g., Bernstein 2000; Rata 2012, 2021), cognitive theory (e.g., Sweller et al. 2019), and evolutionary educational theory (Geary 2005), we theorise academic education as the type of context-independent knowledge that is developed in the sciences and other disciplinary knowledge and is required for the development of secondary abilities (Geary 2005; Sweller et al. 2019). It is the knowledge of the school subjects of mathematics, science, languages, history, geography, music, and so on. In contrast, basic education is context-dependent knowledge and focuses on everyday experiences and acquiring skills to manage these experiences. The volunteer teacher at the community learning centre teaching the refugee children how to ride the Bangkok train system is an example of this type of knowledge.

The context-independent nature of academic education is particularly significant for contemporary refugees. This type of education is generalisable. It can be transferred to other locations, including the country of origin. In the case of repatriation, the Pakistani refugee teenagers can re-enter the public education system by taking the Pakistani Board of Intermediate and Secondary Education examination in academic subjects such as English, physics, chemistry, and mathematics. Basic literacy education, given its context-dependent character, cannot be transferred to future locations, including the country of origin and Western countries.

Refugees' aspirations for an academic education in order to secure professional employment and move from a refugee status to that of a professional immigrant are severely constrained by the reality of global labour markets. The requirements of capitalism for

certain types of labour and in certain quantities are forces that operate at the global level but have effects for refugees at the local level (Bloch 1999).

However, a case can be made to support the refugee parents' aspirations for academic education for their children. A pessimist view suggests that the academic education will not make much difference in the refugees' lives. However, a Gramscian Marxist view or a liberal humanistic view of education justifies academic or intellectual education as providing the means by which the marginalised groups in the world can understand and challenge the politics that maintain inequalities and reproduce their extra-national refugee status.

## 16. Conclusions

Our purpose has been to explain how the 'temporary but permanent' situation that refugees, specifically the Pakistani Christian refugees in Thailand, find themselves in has reactivated the older patron–client relationship. We have argued that the use of patron–client theory is useful in understanding the reactivation of this pre-modern form of sociopolitical relationship. It provides an explanation of the ambiguous co-dependency that operates between INGOs and refugees and sets in place an ongoing unequal relation. We recommend that the INGOs address the unequal relations. This will require an awareness of the inherent inequality of the power structures in the co-dependency relations. Indeed, our intention has been to draw attention to this structural problem in order to encourage such an awareness. Welcoming refugees into decision-making meetings and making decisions 'with them rather than for them' could disrupt the critical components of clientelism. Moreover, establishing accountability and transparency structures and regulations, analogous to the nation-state system, in INGO–refugee relations may create an equilibrium between both parties.

**Author Contributions:** Conceptualization, S.A. and E.R.; methodology, S.A. and E.R.; validation, S.A. and E.R.; formal analysis, S.A. and E.R.; investigation, S.A. and E.R.; resources, S.A. and E.R.; data curation, S.A. and E.R.; writing—original draft preparation, S.A.; writing—review and editing, E.R.; visualization, S.A. and E.R.; supervision, S.A. and E.R.; project administration, S.A. and E.R.; funding acquisition, S.A. and E.R. All authors have read and agreed to the published version of the manuscript.

**Funding:** This research received no external funding.

**Institutional Review Board Statement:** The University of Auckland Human Participants Ethics Committee approved this research project (Reference number: 018666).

**Informed Consent Statement:** Informed consent was obtained from all subjects involved in the study.

**Data Availability Statement:** Data is available on request.

**Conflicts of Interest:** The authors declare no conflict of interest.

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
