# Peer review of "Refugee Education under International NGOs: A Major Shift from National Institutions to Patron–Client Relations"

_socsci, doi:10.3390/socsci11110494_

Round 1
Reviewer 1 Report
This is an interesting argument, and it is a population that merits additional research. The background history beginning around line 150, as well as 180-215 provide good contextualisation. Other important inclusions are quotes such as in lines 353-356 that meetings were for but not with refugees present and the observation that brokers' children were enabled to attend international schools, but not other refugee children.
However, there are many issues with the manuscript. To begin, the phrase "helpless refugees" is highly problematic, as refugees are. highly resilient. It is problematic to view them primarily as helpless victims.
The authors states that a realist conceptual method is deeper than what can be found using empirical data but do not explain HOW. Additionally, the realist method seems to get dropped in favour of a patron-client theory.
There is a disconnect in the opening paragraph which begins by talking about patron-client concepts then moves to a question about helpless refugees without a connect between the two. It comes back up around line 83 but doesn't make sense before that.
Legally, though perhaps not practically, refugees have access to national education systems in Thailand, in contrast to what is stated in lines 99-100. This concept is more contextualised, but not until lines 4536-447.
There are some problematic generalisations, such as statements that refugees were welcomed after WWII. Some were, but some absolutely were not (such as Jews). This should be contextualised.
It would be good to provide references that INGOs have no accountability to nation-state regulations - this seems unusual.
Lines 392-394 indicate that refugees are not provided with adequate food from the INGOs. Can it be demonstrated that this is on purpose, or is it because, as is often the case, the INGOs do not have sufficient provisions?
Line 457 states that INGOs only have "control and power to provide basic literacy education". If so, should they be faulted for doing that? Later there are statements that the INGOs only used untrained volunteers who could not provide more, rather than certified teachers. There needs to be a clarification on whether the INGOs were doing what they could with small funding, or, as implied by the manuscript, they intentionally chose not to use resources that would provide higher levels of education.
The paragraph beginning at line 494 describes using basic education as a means of encouraging repatriation. Unfortunately, only a tin percentage of refugees will ever have the opportunity to resettle in a developed country. As such, it seems problematic to blame the UNHCR for advocating for repatriation when it is safe to do so. Later in the manuscript (lines 599-600), the authors do note that resettlement is unlikely.
Lines 524-528 describe untrained volunteers ignoring refugee requests. Is this intentional, or is it because they simply do not have the training and experience to know how to respond?
Lines 628-633 indicate that academic education would be important for repatriation. Earlier, it is mentioned that this is a goal of UNHCR - so why would the INGOs not be persuaded to provide it?
There are a number of problematic sentence structures. One is at 617-621.
Although this is an interesting research project, I cannot tell from the manuscript whether or not you could address the problems described with the present research.
Author Response
Thank you for your review; please see the attached document.

Reviewer 2 Report
Great topic and very important. Excellent work situating the study in the methodological and theoretical literature. One spot I'd suggest taming is "Ethical issues were a serious concern in our Thai study because Pakistani Chris- 70 tian refugees are the most vulnerable community in Bangkok and face harassment, 71 discrimination, and exploitation. "
This claim is difficult to justify and perhaps shift to '...are one of the most vulnerable..."
Excellent use of the patron-client framework which is very enlightening. Might race and religion also play a role, either intersecting with this patron-client framework, or operating independently?
Could add more distinction to what the Rohingya experienced in Thailand. Is the Christian element a piece that adds complexity to the Pakistani Christian refugee situation?
Figure 1 has some alignment issues.
Powerful ethnographic quotes throughout.
Conclusion could be more robust with, perhaps, recommendations to undo the patron-client differential established by the INGO ecosystem.
Some spacing inconsistency after periods within paragraphs.
Author Response

(The authors gave the same response as above.)

Round 2
Reviewer 1 Report
I find that the authors have responded adequately to my concerns, so I am fine to recommend that the manuscript be published.